# The Control of Postharvest Soft Rot Caused by *Rhizopus stolonifer* on Kokei No. 14 Organic Sweet Potato Roots by Carvacrol, Thymol, and Thyme Oil

**DOI:** 10.3390/foods14071273

**Published:** 2025-04-05

**Authors:** Guangwei Wu, Chenqi Fan, Xueqian Zang, Bei Wang, Yanli Chen, Jingjing Kou, Guopeng Zhu

**Affiliations:** 1School of Breeding and Multiplication, Sanya Institute of Breeding and Multiplication, Hainan University, Sanya 572025, China; 23210902000011@hainanu.edu.cn (G.W.); m13180513601@163.com (C.F.); zangxueqian0117@126.com (X.Z.); 18976583639@163.com (B.W.); chenyanli@hainanu.edu.cn (Y.C.); 2Key Laboratory for Quality Regulation of Tropical Horticultural Crops of Hainan Province, School of Tropical Agriculture and Forestry, Hainan University, Haikou 570228, China

**Keywords:** Kokei No. 14 sweet potato (*Ipomoea batatas [L.]. Lam*.), organic, *Rhizopus stolonifer* (*R. stolonifer*), essential oils (EOs), carvacrol, thymol, thyme oil, postharvest disease

## Abstract

Soft rotting caused by *Rhizopus stolonifer* is one of the most important postharvest decays in Kokei No. 14 organic sweet potato roots. While various methods have been explored for controlling this pathogen, there remains a need for effective, safe, and applicable alternatives, particularly using essential oils (EOs). This study evaluated the efficacy of EOs, specifically carvacrol, thymol, and thyme oil, in controlling *Rhizopus* soft rot. We conducted both in vitro and in vivo tests to assess their effects on fungal mycelial growth, spore germination, and the incidence and severity of soft rot in sweet potatoes, along with quality evaluations of the roots. The results indicated that the vapor phase of carvacrol, thymol, and thyme oil was more effective than the contact phase in inhibiting fungal growth and spore germination. In vivo tests revealed that all three EOs significantly reduced the incidence and severity of soft rot, with thymol and thyme oil at 300 mg/L, and carvacrol at 500 mg/L being the most effective. Quality assessments showed minimal impact on properties such as firmness, weight loss, color, starch, carotenoids, and flavonoids, although residual odors increased. GC/MS analysis confirmed that thyme oil contained high levels of both thymol and carvacrol, along with other antimicrobial compounds, suggesting that the cumulative activity of these volatile compounds enhanced their bacteriostatic effects. Thyme oil demonstrated greater efficacy in reducing soft rot development compared to its individual components, making it a promising biofumigant for controlling postharvest diseases in Kokei No. 14 organic sweet potato roots. These findings emphasized the potential for using thyme oil as a safe and effective approach to managing postharvest decay.

## 1. Introduction

Sweet potato (*Ipomoea batatas [L.]. Lam*.) is considered a major global food crop, with extensive production and consumption in China, which accounts for approximately 54.19% of the world’s total production [1]. The Kokei No. 14 organic sweet potato (*Ipomoea batatas [L.]. Lam*. cv. Kokei No. 14) cultivar performs well in diverse climatic conditions, particularly in the warm, humid environments of Hainan Province, with a 2023 annual value of CNY 0.52 billion (approximately 72.2 million USD). Kokei No. 14 is a nutritious organic sweet potato cultivar rich in starch, carotenoids, and flavonoids but highly perishable, especially during the fresh postharvest phase, making it susceptible to disease even under ideal conditions at 10–15 °C [2,3]. One of the major problems faced by Kokei No. 14 organic sweet potato producers is serious economic losses during long-distance transportation, storage, and retail displays caused by pathogenic fungi such as *Fusarium oxysporum* and *Rhizopus stolonifer*. In addition, numerous characteristics and factors compared to other sweet potato cultivars are attributed to considerable losses of Kokei No. 14 root, such as extremely thin skin that can be easily ruptured, optimal water activity, and high sugar levels that promote fungal growth [4,5]. Postharvest root decay can be caused by several fungal pathogens. Dry rot caused by *F. oxysporum* and soft rot (*Rhizopus* rot) caused by *R. stolonifer* are the two major postharvest decays in roots, which can also cause severe losses in many other vegetables, fruits, and ornamental crops [6,7]. In our current research project, a postharvest disease survey of Kokei No. 14 organic sweet potato roots in Hainan province from 2022 to 2024 indicates that *Rhizopus* soft rot is the second most frequently occurring root decay after *F. oxysporum* decay (unpublished data).

In the effort to reduce postharvest losses caused by *F. oxysporum* and *R. stolonifer*, numerous methods and approaches ere to: (1) evaluathave been employed, and the common ones maintain roots at low temperatures (10–15 °C) to reduce respiration rate during root storage, transit, and retail displays, or fungicide applications during the harvesting stage. Fungicide applications to control soft rot and other diseases during the harvesting stage of sweet potato root production are common practices, but during the postharvest stage, synthetic fungicide applications on roots are no longer allowed or prohibited. Major postharvest pathogens have significantly increased their resistance to synthetic fungicides due to widespread and prolonged agricultural use. Currently, postharvest root decay problems are primarily managed by curing, maintaining the cold chain, and occasionally, by the use of high voltage alternating electric field, chlorine dioxide, or a nitric oxide-modified atmosphere package [8,9,10,11]. However, these treatments do not kill the fungal pathogens, but simply slow or delay disease development. Obviously, the new safe, applicable, effective, and biodegradable postharvest decay control products, methods, and approaches on organic sweet potato roots are constantly needed and being explored. Natural plant products such as essential oils, which are considered biodegradable and less hazardous, could be developed as promising alternatives to synthetic chemicals in controlling pathogenic fungi on vegetables and fruits. As such, they are steadily gaining more attention [12].

Essential oils (EOs) and a considerable number of their components are classified as GRAS (generally regarded as safe) compounds, which could be bioactive alternatives for fungal disease control because of their relatively safe status, wide acceptance by consumers, and the exploitation for potential multi-purpose functional use [13,14]. EOs are secondary metabolites produced by aromatic plants [15]. They are normally volatile and composed of a mixture of terpenes, aldehydes, terpenoids, and alcohols, which confer antimicrobial properties to the EOs [16,17]. The antimicrobial properties of EOs against various fungal pathogens have been well demonstrated [17,18,19]. In order to develop EOs-based safe and applicable approaches to control postharvest soft rotting caused by *R. stolonifer* on ‘Kokei No. 14’ organic sweet potato roots, several different EOs from different sources have been in vitro screened for their activities against *R. stolonifer* in our initial stage of the current study. The EOs that were screened included carvacrol, citral, eugenol, L-carvone, thymol, trans-cinamaldehyde, vanilla, oregano oil, lemongrass oil, spearmint oil, and thyme oil. The preliminary screening study results showed that carvacrol, thymol, and thyme oil were the most effective EOs against *R. stolonifer*. Therefore, these three EOs were selected and focused on in this study (Appendix A).

Thyme oil is a crude EO extracted from *Thymus vulgaris*. Carvacrol (2-methyl-5-(1-methylethyl) phenol) and thymol (2-methyl-5-(1-methylethyl) phenol) isomeric with carvacrol) are the individual components of essential oils. Carvacrol and thymol are among the main components present in thyme oil and some other EOs [20,21]. Carvacrol, thymol, and thyme oils have been reported to have antifungal activities against several postharvest fungal pathogens such as *Alternaria alternata* [22,23], *Botrytis cinerea* [24,25], *Colletotrichum gloeosporioides* [26,27,28], *Colletotrichum musae* [29], *Penicillium digitatum* [30], and *R. stolonifer* [31,32]. Although thyme oil, carvacrol, and thymol have been studied and demonstrated good antimicrobial activities in vitro and decay control effects on banana, blueberry, and pitaya fruit [29,33,34], the information is still limited, especially in research of EO applications on inoculated, naturally infected, and organic sweet potato roots. A limited number of commercial applications of EO-based products have been developed for postharvest organic sweet potato roots.

The objectives of this study were as follows: (1) to evaluate the in vitro activities of carvacrol, thymol, and thyme oil in the volatile and contact phases against the mycelial growth and spore germination of *Rhizopus stolonifer*; (2) observe the morphological changes in carvacrol, thymol, and thyme oil against *R. stolonifer* mycelia and spores; (3) evaluate the effects of carvacrol, thymol, and thyme oil in vivo for the control of *Rhizopus* soft rot using both artificially inoculated and naturally infected Kokei No. 14 organic sweet potato roots; and (4) investigate the influence of carvacrol, thymol, and thyme oil on the physical and chemical quality of root parameters.

## 2. Materials and Methods

### 2.1. Essential Oils, Pathogens, and Roots

Thymol, carvacrol, and thyme oil were purchased from Sigma-Aldrich Corporation (St. Louis, MO, USA). The fungal pathogen *Rhizopus stolonifer* was isolated from diseased organic sweet potato roots produced in Danzhou of Hainan province. The identification of this pathogen was performed and confirmed using both morphological and molecular methods, and pathogenic tests. Morphological identification was conducted by examining key fungal structures, including sporangia, spores, sporangiophores, rhizoids, and stolons. Colony morphology was also observed to characterize the pathogen [35]. For microscopic analysis, fungal hyphae and spores of *Rhizopus stolonifer* were mounted on glass slides and examined under a light microscope. Molecular identification was performed using a polymerase chain reaction (PCR) with the appropriate amplification protocols, followed by DNA sequencing [36]. The obtained sequences were analyzed using NCBI-BLAST (https://www.ncbi.nlm.nih.gov/, accessed on 14 December 2023) to compare homology with sequences in the GenBank database. Strains with high sequence similarity were selected for further analysis. A phylogenetic tree was constructed using MEGA 11 software (version 11.0.3). Fungal identification was achieved using the ITS1F/ITS4 primer set [37]. The fungal culture was maintained on a potato dextrose agar (PDA) at 4 °C until it was used for the experiments. The fresh-harvested organic sweet potato roots (Kokei No. 14 cultivar) used in the experiments were obtained and produced at different times from growers in Danzhou (latitude 19°31′ N, longitude 109°34′ E). Fresh, organic sweet potato roots were used in the planned tests immediately or stored at 10–15 °C, and then they were used the next day.

### 2.2. Analysis of the Volatile Components of Thyme Oil Using the Gas Chromatography–Mass Spectrometry (GC-MS) System

A 1 μL aliquot of the thyme oil’s (Sigma-Aldrich W306509, St. Louis, MO, USA) composition was analyzed using gas chromatography–mass spectrometry (GC-MS, Model 6890, Agilent, Santa Clara, CA, USA). Helium was used as the carrier gas, with a split ratio of 99:1, at a flow rate of 1.2 mL/min. The inlet, ionizing source, and transfer line were kept at 250, 230, and 280 °C, respectively. The oven temperature program was as follows: The oven was kept at 40 °C for the initial 0.5 min; then, the temperature was ramped up to 4 °C/min to reach 250 °C, then increased to 260 °C at a rate of 100 °C/min, and kept at 260 °C for 10 min. The MS was operated in an electron ionization mode at 70 eV, obtaining spectra with a range of 30−250 m/z. Volatile compounds were identified by comparing their mass spectra with library entries [38].

### 2.3. Determining the Contact- and Vapor-Phase Activities of EOs Against R. stolonifer, Mycelial Growth, and Spore Germination

For the determination of mycelial growth in the contact phase, EOs were dispersed as an emulsion in water using Tween 80 (5% *v*/*v*); then, they were added to the autoclaved PDA immediately before it was poured into the Petri dishes (90 × 20 mm). The final concentrations of the EOs in the PDA were set at 50, 100, 200, and 500 mg/L, respectively. In the volatile phase, the plates (90 × 20 mm) containing 20 mL of the PDA, which provides 50 mL of air space, were used in the tests. A sterile filter paper disk (20 mm in diameter, Whatman No. 1) was placed on the inner surface of each plate lid. Either the carvacrol, thymol, or thyme oil was placed on the paper disks to create concentrations of 5, 10, 20, and 50 μL/L, respectively. Prior to each test, *R. stolonifer* was grown on the PDA at 25 °C for one day, and the PDA plugs (7 mm) with fungal mycelia cut from the edge of *R. stolonifer* colonies were transferred to the center of the plates. Four plates were used for each EO concentration. The plates were incubated at 25 °C after they were sealed with parafilm. Two days after incubation, the diameters of fungal radial growth were recorded. The percentage of inhibition by each EO at each concentration level was calculated. For the spore germination determination, in the contact phase, PDA amended with EOs at various concentrations (50, 100, 200, 300, and 500 mg/L) was prepared as described above. In the volatile phase, the concentrations of EO were made at 5, 10, 20, and 50 mg/L, respectively. *R. stolonifer* was grown on PDA for three days at 25 °C to produce sporangiospores. The spore concentration in the suspension was adjusted to 10^5^/mL for germination tests. After the plates were incubated at 25 °C for at least 6 h, the spore germination rates from different treatments were counted with the aid of a microscope. Percentage inhibition by EOs was calculated. Four plates were used for each treatment. The experiment was repeated three times.

During the in vitro testing of the antifungal activities of EOs against *R. stolonifer*, EOs at various concentrations were able to completely stop the mycelial growth of *R. stolonifer*, but it was unclear whether the fungus was completely killed by the EOs at those concentration levels. To verify the fungistatic or fungicidal effects of EOs on *R. stolonifer*, the fungus that completely stopped growing under the various EO treatments was transferred onto new PDA plates to allow it to grow for up to three days at 25 °C. The activity of an EO treatment at a given concentration was considered fungistatic if the fungal growth was observed after the additional incubation period, and it was considered fungicidal if no new growth was detected. The residual fungal growth was monitored by measuring the radial growth of the fungus.

### 2.4. The Effects of Carvacrol, Thymol, and Thyme Oil Vapors on the Morphological Changes in R. stolonifer

After *R. stolonifer* was grown for one day at 25 °C, the fungal PDA plugs were cut from the edge of fungal colonies and transferred onto the empty plates. Four fungal PDA plugs were placed on the bottom of each plate. A small amount (0.75 μL) of carvacrol, thymol, or thyme oil was loaded onto a 2 cm filter paper disk, which was fastened to the inner side of the plate lid. The amount of EO in a plate was 10 mg/L, based on the volume of the plate. The plates were sealed tightly with parafilm and reversely incubated at 25 °C for 12 h. The morphological changes in the fungal mycelia and spores were observed under a stereomicroscope (Nikon SMZ800N, Tokyo, Japan, ×45 magnification), a biomicroscope (Mingmei ML31, Guangzhou, China), and scanning electron microscopy (SEM, Hitachi Regulus 8100, Tokyo, Japan). For scanning electron microscopy, the samples were fixed with glutaraldehyde (2.5%) at 4 °C for 12 h; the mycelia were dehydrated via a graded ethanol series and subsequently coated with gold–palladium. Pictures reflecting observations and changes were taken. The multiple magnifications and imaging parameters were used, including the voltage of 3 kV, the objective aperture of 200 µm, and the working distance of 8.6 mm.

### 2.5. Effects of Carvacrol, Thymol, and Thyme Oil Vapors on the Control of Rhizopus Rot on Artificially Inoculated Roots

Freshly harvested organic sweet potato roots were sorted to ensure uniformity in size, shape, and color, as well as the absence of damage and decay. The roots were surface disinfected with a 0.03% chlorine solution, rinsed with sterile distilled water, and then air-dried at room temperature (22 ± 1 °C). The roots were cut into sections of 5–6 cm in length, and 12 sections were placed at the bottom of a 3.5 L plastic container. Each section was wounded with a sterile metal probe (5 mm × 10 mm) at the center location of the incision. One wound was made for each section.

The spore suspension’s concentration was adjusted to 108 cells per mL^−1^ using a hemocytometer (177- 112C, Watson Bio Lab, Kobe, Japan). In total, 40 μL of the fungal spore suspension was placed onto each wound site. Sterile water was placed in the wound sites to serve as controls. The sachets (7 × 9 cm) were made with a Miracloth (Calbiochem-Behring, La Jolla, CA, USA) containing 60 cm^2^ filter paper (Whatman #1, Whatman International Ltd, Maidstone, England). The concentration of each EO was made at 100, 300, and 500 mg/L, and loaded onto the filter paper in the sachets based on the volume. Sachets containing EO were fastened onto the inner side of the lid of each container. Each container had a small opening hole (2 mm in diameter) to prevent CO_2_ accumulation and O_2_ depletion in the atmosphere. The controls were not treated with any EOs. Each treatment had at least three replicates. Inoculated roots in the containers were incubated for 60 h at room temperature (22 ± 1 °C), and disease incidence and severity were recorded.

The disease severity index was rated on a scale of 0 to 4 based on the percentage decay on the root surface (0 = no decay, 1 = ≤ 25%, 2 = 25–< 50%, 3 = 50–< 75%, and 4 = ≥75% of decay on root surface, respectively). The disease incidence was expressed as a percentage of the diseased root number over the total assessed root number, and disease severity was expressed as a continuous range, from 0 to 100 using the formula [39] as 100 [(0n0 + 1n1 + 2n2 + 3n3 + 4n4)/4n] to convert the disease index obtained from various scales as described above, where n1 to n4 were the numbers of roots in the decay scale of 1 to 4, and n was the total number of roots assessed. The experiment was repeated three times.

### 2.6. The Effects of Carvacrol, Thymol, and Thyme Oil Vapors on the Control of Rhizopus Soft Rot on Naturally Infected Roots

The procedures using naturally infected roots were similar to those of the tests; the same Eos were used to artificially inoculate the roots as described above, except that the roots were not surface disinfected with chlorine and not artificially inoculated with *R. stolonifer*. The whole roots were placed at the bottom of plastic containers equipped with ventilation fans. The vapor concentrations of carvacrol, thymol, and thyme oil used in the tests were 100, 300, and 500 mg/L based on the volume. The sachets containing EOs were secured to the inner side of the container lid, as previously described in artificially inoculated roots. At least three replicates with ten roots for each treatment were used in the tests. The sachets with EO were removed after seven days of root incubation, and the roots were subsequently incubated at 22 ± 1 °C and 85 ± 5% relative humidity. The roots were considered infected when a visible softening of the lesion and/or *Rhizopus* fungal mass were observed. Disease incidence was recorded at 21 d, and disease severity was calculated based on the disease index, on a scale from 0 to 4, as described in the above section. The number of sprouted roots and sprout lengths were also recorded. The experiment was repeated three times.

### 2.7. The Effects of Carvacrol, Thymol, and Thyme Oil Vapors on the Quality of Sweet Potato Roots

#### 2.7.1. The Evaluation of Root Weight Loss and Color Change

Weight loss and external peel color changes over the experimental period were evaluated after 7 d. Differences in weight loss and the external color of roots were measured with the same roots (*n* = 24) before and after treatments. Weight loss was expressed as a loss percentage of the initial total weight [40]. L* (lightness), a* (redness), and b* (yellowness) values of the roots were measured by taking three readings from different equatorial points on each root using a colorimeter (CR-10 Plus, Konica Minolta Optics Inc., Tokyo, Japan), which was used to calculate the chroma value [C* = (a*2 + b*2)^1/2^] and hue angle value [h* = arctan (b*/a*)], which indicate the intensity of color saturation (dull to vivid) and color changes, such as red–purple (0°), yellow (90°), bluish–green (180°), and blue (270°), respectively [41].

#### 2.7.2. Evaluation of Root Firmness, Starch, Carotenoids, and Flavonoids

During the test, firmness, starch, carotenoids, and flavonoid contents before and after treatments were evaluated after seven days of storage. Root firmness was measured using an FTC Texture Analyzer (TMS-PRO, Food Technology Corporation, FTC, Sterling, VA, USA) by measuring the force required for a 2 mm probe to penetrate 20 mm into root flesh. The other parameters were as follows: pre-test speed of 30 mm/s, test speed of 50 mm/s, post-test speed of 30 mm/s, and trigger force of 0.375 N. The probe always returned to the trigger point before beginning the next cycle [42]. The values were expressed as a (N). At day 0, a group of fresh roots was measured at the beginning of the assay in order to evaluate the change in firmness over time. Firmness was calculated with the measurements along the equatorial zone of each root by using the maximum peak force of the first compression. For the texture profile analysis (TPA), the roots were peeled, and a disk (2 × 1 cm) was cut in the middle of the whole root. A two-cycle compression test was performed on the equatorial part of the disk using an aluminum cylinder probe (75 mm diameter), which was used to compress the samples to 10% of their original thickness at a compression rate of 30 mm/s. Three measurements were performed for each root, and each treatment was measured with three different roots.

For the nutrient determination, flesh tissues were taken from the end and middle locations of each root after the peel was removed. The samples were cut into pieces, frozen in liquid nitrogen immediately, and ground into a fine powder using a grinder (IKA A11 basic, Staufen, Germany); then, the samples were stored at −80 °C. Starch, carotenoid, and flavonoid content analyses were measured by the UV-Vis spectrophotometer (SP-756 P, Shanghai Spectrum Instruments Co., Shanghai, China) using the starch content assay kit (BC0700, Solarbio, Beijing, China) [43], carotenoids content assay kit (BC4335, Solarbio, Beijing, China) [44], and flavonoids content assay kit (BC1330, Solarbio, Beijing, China) [45]. Three replicates were conducted for each treatment.

#### 2.7.3. Odor Analysis by Electronic Nose and Sensory Panel Test

The headspace aroma of EO treatments was traced by the 18 sensor reactions of E-nose (ISENSO INTELLIGENT, Shanghai, China). The names (SN-1~SN-18) and corresponding response substances of the 18 metal-oxide semiconductor chemical sensors were listed in Appendix A. In total, 5 g of diced root flesh (from day 0 and day 7 after 14 d of storage) was placed in a 40 mL specialized headspace bottle, then capped and left at constant room temperature for 30 min to allow volatiles to disperse fully. Before the first measurement, the E-nose device was cleaned for 1 h by injecting reference air into the gas path to normalize the sensors. The constant gas flow rate in headspace was set to 1.0 L/min, and the test phase lasted for 120 s. Between the time intervals for every two measurements, the device was cleaned for 180 s until it returned to baseline. Five replicates were carried out for each sample.

In the sensory panel experiment, 25 carefully selected and experienced panelists, including students and staff members, conducted evaluations at the Key Laboratory for Tropical Horticultural Crop Quality Management. The methodology followed the approach described in reference [46]. The Society of Sensory Professionals (SSP) recommends four or more panelists; it provided their training to justify lower numbers. The recruited panelists ranged in age from 20 to 56 years, with a gender distribution of 52% male and 48% female. Each panelist underwent extensive training and participated in the sensory evaluation trial after providing informed consent. The sensory assessments were conducted in a well-lit, well-ventilated evaluation room, which was maintained at 22 ± 1 °C, using individual tables with privacy dividers. Steamed sweet potato samples with skin (50–100 g) were meticulously prepared and presented in 50 mL disposable plastic containers, each labeled with a unique, randomly assigned three-digit code to ensure sample anonymity and minimize bias [47]. Panelists evaluated the samples based on a predefined list of sensory descriptors, including appearance (AR), flesh color (FC), firmness (FM), sweet potato odor (SO), off-odor (OO), sweetness (SW), fibrousness (FB), viscosity (VS), off-flavor (OF), and overall acceptability (OA) (Appendix A). The evaluations used a 9-point hedonic scale, ranging from 1 (extremely poor) to 10 (extremely good). The sample presentation order was randomized for each panelist, and specific information regarding the steamed sweet potatoes with skin was retained during the evaluation. During the sensory assessment, panelists recorded their ratings for each attribute on standardized scoring sheets designed to capture quantitative scores and qualitative comments, providing a comprehensive profile of the sensory characteristics of each sample. The collected data were subsequently subjected to statistical analysis to determine significant differences between treated and control samples, and to explore potential correlations among various sensory attributes [48].

### 2.8. Data Analysis

The same experiments were conducted in both 2023 and 2024 with consistent results, and the data from 2023 were presented in this paper. Analysis of variance of data were performed using the IBM SPSS Statistics (Version 20), GraphPad Prism 10.0, and OriginPro 2021 (OriginLab^®^, Northampton, MA, USA). Descriptive statistics, including mean scores and standard deviations, which were calculated for each attribute, and analysis of variance (ANOVA) was performed with Duncan’s multiple range test (*p* ≤ 0.05) for treatment means comparison. Principal component analysis (PCA) and graphical representations were conducted, and the results were displayed as 3D plots.

## 3. Results and Discussion

### 3.1. Component Analysis of Thyme Oil Using GC-MS System

The chemical composition of the commercial thyme oil was determined using a GC-MS system. Table 1 shows that in this thyme oil, a total of 22 volatile compounds were identified. Most volatiles detected in this study were consistent with those of previously published studies [22,23,49,50]. The major volatile compound present in the thyme oil used in our current study was thymol, with p-cymene, γ-terpinene, and linalool. Carvacrol is also the most abundant compound, which is congruent with previously published thyme oil profiles [22,50]. Thymol, as the major component of the commercial thyme oil, had concentrations varying between 45.7 and 75.4%; the second most important constituent was carvacrol, which is an isomer of thymol with a strong antifungal function, which has also been frequently observed as another compound (4.9 to 5.4%) in the composition of thyme oil. γ-terpinene, p-cymene, and linalool were the other main compounds detected in commercial thyme oil, and p-Cymene was previously found as a precursor of the biosynthesis of thymol and carvacrol in thyme oil [51].

### 3.2. Antifungal Effects of EOs on the Mycelial Growth and Spore Germination for R. stolonifer Under In Vitro Conditions

#### 3.2.1. Isolation and Identification of R. stolonifer

Morphological analysis revealed that colonies on PDA were white, dry, and opaque, with long, filamentous hyphae (Figure 1). Aerial stolons extended across the medium surface, giving rise to abundant black spores. The sequences obtained using the primer set ITS1F/ITS4 were identified through BLAST analysis against the NCBI database. Phylogenetic tree construction revealed that the pathogen was clustered with other Rhizopus species and was closely related to *R. stolonifer*. Therefore, molecular characterization confirmed that the species was *R. stolonifer*.

#### 3.2.2. Preliminary Assays of Selected EOs Against *R. stolonifer* Mycelial Growth In Vitro

The preliminary screening of the essential oils, including seven pure components (thymol, carvacrol, eugenol, citral, trans-cinamaldehyde, vanilla, and L-carvone) and four crude extracts (thyme oil, oregano oil, lemongrass oil, and spearmint oil), as mentioned in Appendix A, showed different inhibitory effects on the target pathogen *R. stolonifer* using both medium direct contact and vapor-phase evaluations. Thymol, carvacrol, and thyme oil demonstrated significantly greater effectiveness against *R. stolonifer* compared to the other EOs tested under our conditions. Therefore, thymol, carvacrol, and thyme oil were selected and spotlighted for further investigation as safe and alternative approaches for *Rhizopus* soft rot control on organic sweet potato roots.

#### 3.2.3. Antifungal Activities of Carvacrol, Thymol, and Thyme Oil Against *R. stolonifer* Mycelial Growth In Vitro

The antifungal activities of different amounts of tested EOs studied using contact and volatile assay methods for *R. stolonifer* in vitro are shown in Figure 2. The obtained results revealed that the type of EOs, different application phases, and their various concentrations significantly (*p* < 0.05) affected the mycelial growth of *R. stolonifer* under our test conditions.

All three tested EOs showed that the inhibitory effects were dose-dependent dependent. For contact phase activity, thymol completely (100%) inhibited the mycelial growth at 100 mg/L, while carvacrol and thyme oil were less effective than thymol and required a higher concentration of 200 mg/L to completely inhibit fungal mycelial growth. When the vapor phases of carvacrol, thymol, and thyme oil were subjected to testing against the mycelial growth of *R. stolonifer* in this study, it was found that all three EOs completely inhibited the fungal mycelial growth at a 10 mg/L level. This clearly indicates that the volatile phases of the three tested EOs are more effective than the medium direct contact phases against the *R. stolonifer* mycelial growth.

EOs may inhibit the growth of fungi either temporarily (fungistatic) or permanently (fungicidal). When the complete suppression of *R. stolonifer* by the tested EOs at a setting concentration level was observed, the effects of these three EOs on the viability of *R. stolonifer* on PDA were determined by transferring the fungal plugs to fresh PDA plates to allow the fungus to regrow. If no new mycelial growth inhibition was observed, it indicated that the EOs killed the fungus and exhibited fungicidal effects. Thymol at 100 mg/L in the contact phase treatment completely inhibited the mycelial growth of *R. stolonifer* and exhibited fungistatic activity. However, at the concentration of 200 mg/L, it was demonstrated that all three EOs in the contact phase completely killed *R. stolonifer* and exhibited fungicidal activity. For the vapor phase activity tests of these three EOs against *R. stolonfer*, all three EOs completely inhibited the mycelial growth of *R. stolonifer* at 10 mg/L, but they demonstrated fungistatic activity. As the vapor concentrations increased to 20 mg/L, thymol and thyme oil showed fungicidal activity, but not for carvacrol. The vapor concentration at 50 mg/L of carvacrol showed fungicidal activity and completely killed *R. stolonifer* in our study. A comparison between contact and vapor fungicidal activities of the three EOs against *R. stolonifer* indicates that vapor phase effects were about 5, 5, and 2 times higher than medium contact phase for thymol, thyme oil, and carvacrol, respectively.

Antifungal activities of various EOs in vitro against *R. stolonifer* has also been demonstrated by many other researchers. Sellamuthu, Sivakumar, and Soundy [26] reported that thyme oil at 5 μL in the Petri dish (90 mm in diameter) containing 15 mL of PDA with vapor phase completely inhibited the radial mycelial growth of *R. stolonifer*. The growth of *R. stolonifer* was completely inhibited by thymol and carvacrol at 250 mg/L in the contact phase [52]. However, Plotto et al. [31] reported that the vapors of thyme oil, thymol, and carvacrol at 50 mg/L had a fungicidal effect against *R. stolonifer*, while 500 mg/L of these three EOs were needed in the contact phase to exhibit the same fungicidal effect against *R. stolonifer*. Nabigol and Morshedi [53] reported that carvacrol in the contact phase in PDA exhibited fungistatic and fungicidal effects on the mycelial growth of *R. stolonifer* at 150 and 600 mg/L, respectively, while 300 and 1200 mg/L of thymol were needed to reach the fungistatic and fungicidal effects, respectively, against *R. stolonifer*. In addition, carvacrol at 2 μL per plate (90 mm in diameter) containing 15 mL of PDA in vapor phase completely inhibits the growth of *R. stolonifer* [54]. The results from this study are basically consistent with the reports regarding similar studies.

The results from the current study further support the previous reports that volatile inhibitory and antifungal activities of EOs on mycelial growth were more effective than contact inhibitory effects [22,55,56]. In our study, EO vapors showed complete inhibition against *R. stolonifer* in the range of 10 to 50 mg/L of carvacrol, thymol, and thyme oil, but the higher concentration of 100 to 200 mg/L of the EOs in the contact phase was required to completely inhibit the growth of the fungal mycelia. One of the possible reasons that have been suggested for the vapor phase being more effective than the contact phase is that the lipophilic EO molecules in the aqueous phase associate to form micelles and thus suppress the attachment of the EOs to the organisms, whereas the vapor phase allows free attachment [57]. The better attachment of oil molecules in the vapor phase to the lipophilic fungal mycelia compared to the liquid phase thus allows the oil molecules to exert their antifungal effect directly on fungal mycelia [58].

### 3.3. Antifungal Activities of Carvacrol, Thymol and Thyme Oil Against R. stolonifer Spore Germination In Vitro

As shown in Figure 3A, the spore germination rates of *R. stolonifer* were suppressed by carvacrol, thymol, and thyme oil at all tested concentrations (50, 100, 200, and 500 mg/L) in the contact phase by 55.25% to 100% compared to the control treatments. The spore germination of *R. stolonifer* was inhibited completely by thyme oil at 100 mg/L, and by carvacrol and thymol at 200 mg/L in the contact phase. Thyme oil was significantly more effective than carvacrol or thymol at 50 and 100 mg/L levels, suppressing *R. stolonifer* spore gemination in the contact phase. In vapor phase tests, all three EOs completely inhibited the spore germination of *R. stolonifer* at 20 mg/L, and the suppression effects were similar among the three EOs at 5 and 10 mg/L levels (Figure 3B). By comparing the fungal spore suppression effects between contact and vapor phases of the EOs tested, it was clearly indicated that the vapor phase activities of carvacrol, thymol, and thyme oil were about 10 times higher than those contact phase in suppression of *R. stolonifer* spore germination.

### 3.4. Effects of Carvacrol, Thymol, and Thyme Oil on Morphological Changes in R. stolonifer

After exposure to a concentration of carvacrol, thymol, and thyme oil vapor at 10 mg/L for 12 h, the *R. stolonifer* plugs on PDA exhibited degenerative changes in hyphal morphology compared to the control (Figure 4). The EO-treated fungal hyphae and spores appeared degraded, shriveling, crinkled, and desiccated, while the control was big, thick, elongated, and smooth. Hyphal fragmentation, cytoplasmic coagulations, cell wall disruption, and necrosis on the fungal hyphae were also observed in EO-treated *R. stolonifer*. These observations indicated that EOs at the treated concentration and duration may not completely kill the *R. stolonifer*, but they can significantly alter its physiology and reduce its pathogenicity in causing soft rot on organic sweet potato roots.

Similar alterations and changes were observed in *R. stolonifer* in strawberries when treated by essential oils extracted from leaves of *Lippia sidoides* (49.46% thymol) applied with carboxymethylcellulose coating [59] and in peach vapored by EOs carvacrol and eugenol [54]. The damage effects of EOs on fungal mycelia have been widely proven by many other studies [60,61,62]. It has been suggested that such modifications may be related to the interference of essential oil components such as terpenoids with the cell membrane and result in the loss of integrity, leading to the loss of organelles depleted of cytoplasm [61,62].

Several studies have been conducted to understand the mechanisms of action of EOs. In many cases, the antifungal activity of the essential oils results from their hydrophobicity characteristic, which enables them to partition in the lipids of the bacterial cell membrane and mitochondria [63]. This leads to relaxation of the bacterial cell structures, leakage of ions and other cell contents, which contributes to lysis and death [64]. Thyme oil treatment of *R. stolonifer* might also cause destructive effects on the fungal enzyme systems, including the cell-wall degradation-related enzymes pectolytic enzymes, since in this study, thyme oil-treated *R. stolonifer* grew significantly slower than untreated *R. stolonifer* in a liquid medium with the polygalacturonic acid as the only carbon source. This indicates that the destructive effectiveness of EOs on fungal pathogenicity and virulence-related enzymes could be one of the mechanisms that EOs may have for fungal decay control on roots.

In our current study, the in vitro antifungal activity results showed that the suppressions of these three EOs are similar on mycelial growth and spore germination of *R. stolonifer* (Figure 2 and Figure 3). This indicates that thymol and carvacrol might significantly contribute to the antifungal activity of thyme oil against *R. stolonifer*. However, the EO’s antifungal activity may not be only attributed to its major components, and minor compounds or synergism effects among compounds present in the EOs could also contribute to the antifungal activity [65]. Despite the presence of the major compounds identified, 17 other compounds were present in the thyme oil used in the current study, and were considered as minor components. According to previous studies, the minor compounds such as eugenol [66], caryophyllene [67], and α-pinene [68] have also shown antifungal activities. In addition, p-Cymene is reported to potentiate the activity of carvacrol [65]. Nguefack et al. [69] also reported that mixed fractions of EOs showed synergistic effects on antifungal activities.

### 3.5. Effects of Carvacrol, Thymol and Thyme Oil on Rhizopus Soft Rot Development on Artificially Inoculated Organic Sweet Potato Roots

Since the in vitro antifungal effects of carvacrol, thymol and thyme oil in vapor phase were much more effective against both *R. stolonifer* mycelial growth and spore germination than those in contact phase, the volatile phase of the EOs was employed to reduce the fungal infection and disease development on organic sweet potato roots after artificially inoculation with *R. stolonifer*. The vapor forms of the EOs appear to be more practical in actual industry applications. Furthermore, the quite high concentrations of EOs were used for in vivo experiments, since there might be interactions among the compounds of EOs and the food matrix [70]. The results of the effects of thymol, carvacrol, and thyme oil on the *Rhizopus* soft rot control are shown in Figure 5. After inoculation and storage at room temperature (22 ± 1 °C), all control roots developed initially watery lesions which gradually turned into soft rot. The pictures of Figure 5A–C showed that fungal mass amount (mycelia and sporangiophores) was much less on all EO-treated roots than that of control roots. Under various EO treatments, decay incidences significantly reduced in a few cases, but decay severities were significantly reduced in many different treatments (Figure 5D). Carvacrol, thymol, and thyme oil at 500 mg/L, 300 mg/L, and 300 mg/L showed the highest reduction in disease inhibition and severity. Lower EO concentration of 100 mg/L or higher EO rate of 500 mg/L was generally not better than 300 mg/L rate for thymol and thyme oil under the test conditions. Thyme oil treatment exhibited a higher percentage of disease inhibition and severity reduction compared to carvacrol and thymol.

The results from our current study were in agreement with [71,72], who demonstrated the potential of 2-methylbutanoic acid, isobutyric acid, perillaldehyde, salicylaldehyde, and cinnamaldehyde EO components in reducing the growth of *R. stolonifer* on treated organic sweet potato roots. Our findings also support previous studies that emphasized a 65% reduction in disease incidence caused by *R. stolonifer* in papaya when coated with 0.1% thyme oil [73]. Additionally, carvacrol, thymol, and thyme oil have been reported to effectively control *Rhizopus* soft rot disease development on other fresh produce, including strawberries, peaches [74], nectarines [75], and tomatoes [32].

It is also noticeable from our data that, at EO concentrations of 100, 300, and 500 mg/L for vaporing the roots, carvacrol treatment significantly increased the efficacy for disease severity reduction, but thymol and thyme oil did not consistently enhance the efficacy. It suggests that there is a threshold rate level of EO treatment for the root to achieve the maximum decay control. Too low a concentration of EO treatment may not yield satisfactory decay control, but too high a concentration can lead to root injuries and less decay control efficacy. Plotto et al. [31] observed a similar situation for *Rhizopus* rot control using thyme and oregano oil on *Rhizopus* spore-inoculated tomato, and on the contrary, the higher the concentration applied, the more disease developed. They explained that this could be due to a local phytotoxic effect of EOs in the wound, making the issue more susceptible to infection by germinating spores.

*R. stolonifer* is a very aggressive fungal pathogen with rapid growth rates and decay development on organic sweet potato roots under room temperature. The experiment was performed under root wound conditions and an elevated inoculum pressure (4 × 10^6^ spores per wound site inoculation). Therefore, the present assay approaches provided favorable conditions for the development of the disease, especially without the use of low temperature such as the temperature study showed the spores of *R. stolonifer* did not germinate at 5 °C (Figure 5E). The temperature study showed that *R. stolonifer* can grow over a range of 5 to 35 °C, with the optimum growth at 23 to 24 °C. The fungus did not grow when the temperature was lower than 5 °C or higher than 35 °C. *R. stolonifer* is a fast-growing fungus under suitable conditions [32]. The transit and storage of sweet potato roots are normally under cool temperatures where *R. stoloniter* is suppressed for its growth and infections, and soft rot development is at minimal levels. However, *R. stolonifer* infection and soft rot development are only delayed by the low temperature, and it could still be a problem when the temperature rises above 5 °C, such as during market display or in the consumer’s home. The situation was also similar to stone fruit [76,77,78]. We also found that *R. stolonifer* spores did not germinate at 5 °C during 24 h on PDA.

### 3.6. Effects of Carvacrol, Thymol, and Thyme Oil on Rhizopus Soft Rot Control on Naturally Infected Organic Sweet Potato Roots

The effects of carvacrol, thymol, and thyme oil on the control of *Rhizopus* soft rot were further evaluated using organic sweet potato roots under natural infection at room temperature (22 ± 1 °C). Three weeks after incubation, control roots exhibited 100% incidence of *Rhizopus* decay, while treatments with carvacrol, thymol, and thyme oil at concentrations of 100, 300, and 500 mg/L reduced decay incidence by 26.67 ± 5.77 to 100 ± 0% (Figure 6C). No significant difference in decay incidence reduction was observed between the 300 and 500 mg/L concentrations of carvacrol. Vapor treatments with carvacrol, thymol, and thyme oil at 100, 300, and 500 mg/L significantly reduced *Rhizopus* rot severity from 46.67 ± 1.44 to 100 ± 0% (Figure 6C). For thymol treatments, a concentration of 300 mg/L was the most effective; however, the higher concentration of 500 mg/L caused root injuries and reduced decay control efficacy. In contrast, for carvacrol and thyme oil, raising the application rate from 100 to 500 mg/L significantly enhanced their ability to prevent decay (Figure 6C). Among the three EOs tested, thyme oil exhibited significantly greater efficacy than carvacrol and thymol in controlling *Rhizopus* soft rot in non-artificial inoculated sweet potato roots. Comparing these results to the data obtained from the evaluation of EOs for *Rhizopus* decay control on artificially inoculated roots, the trends in decay control were consistent, but more effectiveness was observed on naturally infected roots. This was mainly due to the much higher *Rhizopus* inoculum used on artificially inoculated roots compared to the naturally occurring *Rhizopus* inoculum on non-artificially inoculated roots. In both test types, the increase in carvacrol concentration from 100 to 500 mg/L enhanced decay control efficacy. However, higher concentrations of thymol (500 mg/L) did not improve decay control compared to the lower concentration (300 mg/L) in either test. Thyme oil treatments increased control efficacy in naturally infected roots with rising concentrations (100 to 500 mg/L), though higher concentrations (500 mg/L) led to increased tuber incidence in artificially inoculated roots (Figure 5C and Figure 6C). In addition, both types of evaluation showed that thyme oil was more effective than carvacrol and thymol at the same application levels. This can be largely attributed to the properties and the component mixture of thyme oil. The presence of multiple volatiles, including carvacrol and thymol as the primary compounds, contributed to the antifungal activity and decay control observed. The syngenic effects from the mixture of the active components could also play a role in enhancing efficacy. The syngenetic effects of thymol and carvacrol in combination, compared to their individual EO treatment against Botrytis cinerea, causing gray mold in strawberries, have been reported [79].

Several studies have demonstrated that various EOs such as clove oil [80], cinnamaldehyde oil [72], tea tree oil, lemongrass oil, peppermint oil, and sweet basil oil, have shown good potential for controlling *Rhizopus* soft rot on various vegetables and fruits (yam, peach, tomato) including sweet potato roots using inoculation systems [81,82,83]. However, the tests on the non-inoculated sweet potato root (natural infection) system were rare. Evaluating essential oil efficacy using a natural infection system is more representative of commercial operations. For the commercial transport of sweet potato roots, prompt precooling to temperatures of 10–15 °C and holding at such temperatures in transit, storage, and during marketing will minimize root losses [84]. *Rhizopus* soft rot on various products could be significantly reduced when temperatures are kept below 5 °C [85,86]. However, during the transport chain and postharvest storage of sweet potato roots, some parts, such as cold storage, retail displays, and consumer homes, temperatures may exceed 5 °C, leading to the development of *Rhizopus* decay and potential root losses. EO treatments, such as thyme oil, could provide good potential to control *Rhizopus* soft rot and other decays for sweet potato roots. Additionally, high concentrations of EO treatments (300 and 500 mg/L) significantly inhibited the sprouting of roots. After three weeks of storage, control roots exhibited sprouts (>10 mm), and most of which naturally colonized by mycelium (Figure 6(a–c)). In contrast, the EO-treated roots showed no signs of sprouting (Figure 6(Ba)), and any sprouts that initially appeared turned brown and ultimately died (Figure 6(Bb–d)). Both the numbers and length of sprouts were significantly reduced by EO treatments. Thyme oil demonstrated the lowest sprouting rate across all concentrations compared to carvacrol and thymol, reflecting the cumulative activity of all volatile compounds. Previous studies have shown that various EOs, including carvone [87], jasmine, mint, and clove oils [88], effectively suppressed sprout growth and minimized tuber sprout length and number in sweet potato roots. The EOs used in our current study played a dual role in both controlling *Rhizopus* soft rot and inhibiting sprouting in postharvest sweet potato roots.

### 3.7. Effects of Carvacrol, Thymol, and Thyme Oil on Organic Sweet Potato Root Quality

Table 2, Table 3 and Table 4 indicated that vapor treatment of unwounded sweet potato roots with carvacrol, thymol, and thyme oil at all tested concentrations (100, 300, and 500 mg/L) had no significant impact on physiological and chemical indicators after one week of storage at room temperature (22 ± 1 °C). Weight loss increased linearly for both control and EO-treated roots, with no significant differences observed between groups. Similarly, firmness remained consistent throughout the seven-day incubation period, showing no significant variations. EO treatments did not significantly affect starch concentration, carotenoid content, or flavonoid levels during storage. Overall, starch and carotenoid content increased with storage time, while flavonoid levels decreased. In this study, EO vapors had no significant impact on hue (h°), chroma (C*), and light (L*) values of sweet potato root peels. Roots treated by EOs at concentrations up to 500 mg/L exhibited no phytotoxicity effects on surface color. The results suggest that vapor treatments with carvacrol, thymol, and thyme oil did not interfere with the storage process of sweet potato roots, and no significant differences were observed between treated and untreated roots.

Previous studies have reported beneficial or neutral effects of EOs on horticultural quality parameters after harvest. Abou-Elwafa et al. [88] found that EOs from jasmine, mint, and clove reduced decay in sweet potato roots without impacting weight loss, total carotene, or sugar content. Similar findings were reported for ‘Macaca’ potatoes [89] treated with peppermint oil vapors. Jia and Lyu et al. [90,91] also found that citronella and grape seed EOs had no significant effect on the physicochemical parameters of potato tubers during storage. Thus, it can be concluded that the quality properties of sweet potato roots during natural storage were not negatively affected by the presence of EOs in the vapor phase.

### 3.8. Effects of Carvacrol, Thymol, and Thyme Oil on Organic Sweet Potato Root Odors and Sensory Analysis

After seven days of storage with EO, the odors of uncooked sweet potato roots were analyzed using an E-nose (Figure 7A). At the end of the storage period (7 d), compared with the untreated roots, carvacrol, thymol, and thyme oil may leave a residue on the treated roots and cause unpleasant odors. Higher concentrations (300–500 mg/L) of EOs exhibited greater residual odors, resulting in stronger off-odors, particularly with thyme oil. However, after removing the EO treatment sachets and continuing to store the roots for another seven days with ventilation vans (14 d) at room temperature, the residual odor of all EOs significantly diminished, becoming similar to the control roots. These findings indicate that while EO treatments initially caused residual odors, their rapid volatilization over time mitigated their impact on root odor. Sensory attributes of the sweet potato roots treated with EOs were also evaluated at the end of the storage period (7 d) and one week after EO removal (14 d), as shown in Figure 7B. Similarly to The odor analysis, sensory evaluations revealed significant differences on 7 d compared to the control, particularly in off-odor (OO), off-flavor (OF), and overall assessment (OA), which increased with higher EO concentrations. Also, this off-flavor significantly improved after seven days of continued storage by the volatilization properties of EOs at room temperature. The high concentration EO treatment (500 mg/L) still exhibited a residual taste compared to the control, but no significant sensory differences were observed in roots treated with 100 to 300 mg/L levels.

Principal component analysis (PCA) was conducted using all evaluated physicochemical parameters to highlight the distinct characteristics of sweet potatoes treated with EOs (Figure 8). The cumulative contribution of the principal components exceeded 85%, sufficient to explain the total variance in the data set. Selecting PC1 and PC2 was appropriate for distinguishing sensory attributes among different groups. By the end of storage (7 d), PC1 and PC2 explained over 90.70% of the total variation, with PC1 accounting for 70.40% and PC2 contributing 20.30% (Figure 8a). After the removal of EOs and an additional week of storage (14 d), PC1 and PC2 explained over 99.70% of the total variation (Figure 8b), with PC1 contributing 96.70% and PC2 3.00%. On Day 7, sweet potatoes treated with carvacrol, thymol, and thyme oil showed clear separation from the control group, indicating that the treatments left a strong residual odor, negatively impacting FB, SO, and FG, consistent with lower sensory scores (Appendix A). Roots treated with 500 mg/L thyme oil exhibited distinct separation from other treatments, suggesting that high concentrations of thyme oil led to an undesirable sensory experience. After one week of EO removal (14 d), FB and OF were the primary influencing factors for PC1, and their sensory scores further validated the PCA results. The roots treated with 500 mg/L thyme oil remained distinctly separated from other treatments, while samples treated with 100–300 mg/L clustered with the control group in the PCA plot. This indicated that after one week of volatilization, the sensory characteristics of these treated roots became similar to those of the control group, leading to a noticeable improvement in odor, consistent with the E-nose analysis results.

Sweet potato roots treated with 500 mg/L of EO vapor exhibited undesirable off-odor and off-flavor, potentially limiting their marketability. Wang et al. [92] found that spraying yam with 2.5 μg/mL cinnamon EO resulted in a slight reduction in flavor ratings, limiting its use in fresh markets where aroma and taste are critical. Romeo et al. [93] reported that lemon verbena, cypress, and lemon-balm EOs effectively controlled bacteria in carrots but caused slight off-smells and tastes at 5% concentration, which dissipated over time. Similar results were found in fresh-cut asparagus treated with cinnamon essential oil after 5 d of cold storage [94]. 

The high volatility of EOs, with their price, as well as their odor and impact on flavor, are the most common challenges encountered in the application of EOs in the food industry. Our study indicated that thyme oil was more effective than carvacrol and thymol for *Rhizopus* decay control. Thyme oil is a crude extract of EO, which has a lower price with a better effect than the single components of carvacrol and thymol. The intense sensory attributes of some of these compounds may also be an impediment to their use in fresh commodities. In this study, the effective antifungal effects were achieved with lower doses of thyme oil, minimizing costs while maintaining efficacy. Applying EOs for a short period of seven days just during the early storage phase significantly inhibited *R. stolonifer* infections, and reduced the duration of exposure and potential residual effects associated with prolonged EO treatments. Additionally, direct contact between EOs and sweet potato roots exacerbated the residual odors. The fumigation method employed in this study exhibited better antifungal properties than the direct contact phase, required lower quantities of EOs, reduced direct contact with the root surface, and limited residual taste. Although the employment of EOs had no negative effect on the sweet potato quality, the increasing application of EOs worldwide has raised serious concerns with respect to their eventual adverse health and environmental effects, which are still waiting to be confirmed.

## 4. Conclusions

The evaluation of carvacrol, thymol, and thyme oil as potential new, safe, and effective products for postharvest *Rhizopus* soft rot control on Kokei No. 14 organic sweet potato roots was conducted from preliminary EO selection, in vitro antifungal activity assessments, and in vivo decay control activity using both inoculated and non-inoculated roots. The results indicated that carvacrol, thymol, and thyme oil exhibited strong activity against mycelial growth and spore germination in both contact and vapor phases. Antifungal activity in the vapor phase is more effective than that in the contact form. Carvacrol, thymol, and thyme oil at 300 mg/L in vapors reduced *Rhizopus* decay severity from 76.06% to 95.00% on artificial inoculated roots and non-artificial roots. Thyme oil performed significantly better than carvacrol and thymol. Vapor treatments of carvacrol, thymol, and thyme oil at the tested rates did not affect root weight loss, firmness, coloration, or the levels of starch, carotenoids, and flavonoids, but residual odors were observed at 300 and 500 mg/L rates. Volatile component analysis showed that thymol and carvacrol are among the major compounds in thyme oil. Microscopic observations of mycelium and spores after EO treatment indicated that the EOs may alter the physiology and pathogenicity of the fungal mycelia, ultimately reducing their capacity to induce soft rot in organic sweet potato roots. The results of these studies suggest that thyme oil combined with low temperatures may have good potential to be developed for the applications under commercial Kokei No. 14 organic sweet potato packing and delivery conditions to reduce postharvest root losses and prolong root shelf life. However, further research is needed to evaluate the impact of residual odors on consumer acceptance, explore the physiological changes in fungal mycelia and spores induced by EOs, and provide deeper insights into their antifungal mechanisms. Additionally, the antifungal efficacy of these essential oils should be evaluated across different sweet potato varieties and other crops susceptible to *Rhizopus* soft rot.

## Figures and Tables

**Figure 1 foods-14-01273-f001:**
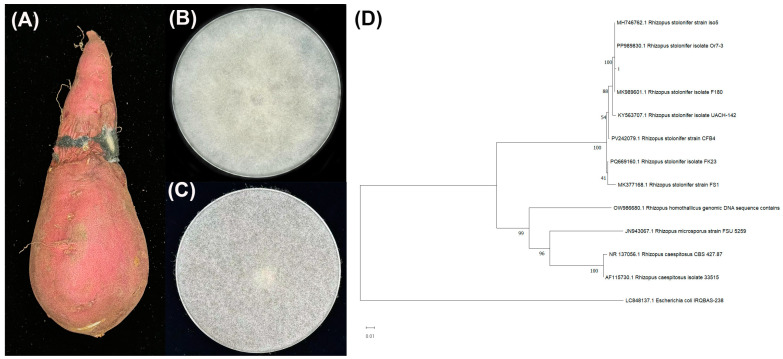
Symptoms of sweet potato postharvest diseases collected from local farms in Danzhou (**A**); *Rhizopus* rot on roots. (**B**) Colonies on PDA plates at 3 and (**C**) 7 days after inoculation. (**D**) The maximum likelihood tree of the *R. stolonifer* isolates as inferred from the combined data sets of the ITS 1 and ITS4 gene sequences constructed by the MEGA 11 program. The tree is rooted to Escherichia coli IRQBAS-238. The numbers on the branches indicate the bootstrap values. The scale bar indicates expected changes per site. The isolates from the present study are indicated by i.

**Figure 2 foods-14-01273-f002:**
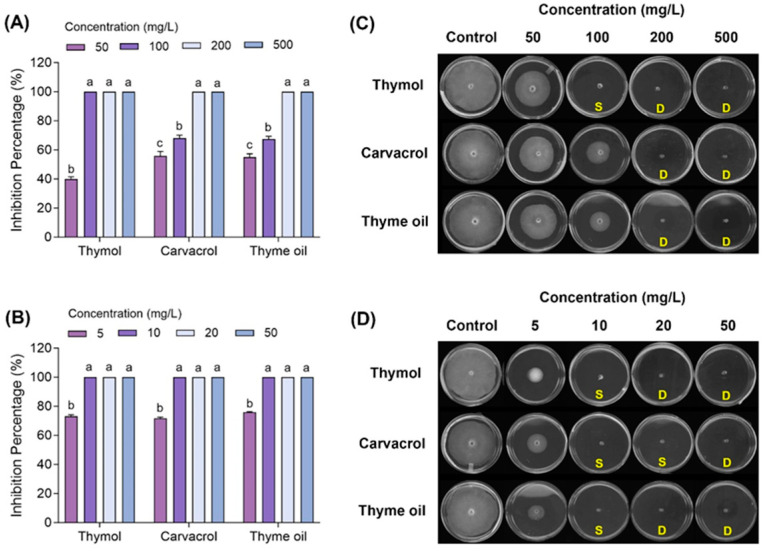
Effects of carvacrol, thymol, and thyme oil in the contact and vapor phases on the mycelial growth of *Rhizopus stolonifer* on the potato dextrose agar (PDA). The viability of *R. stolonifer* was determined 24 h after exposure to essential oils, and it was transferred to a fresh PDA plate; S = fungus did not grow or survive, and D = fungus died. Mycelial growth under the contact phase (**A**,**C**), and volatile phase (**B**,**D**) tests. The means followed by different letters at the same sampling point indicated statistical differences according to Tukey’s test (*p* < 0.05).

**Figure 3 foods-14-01273-f003:**
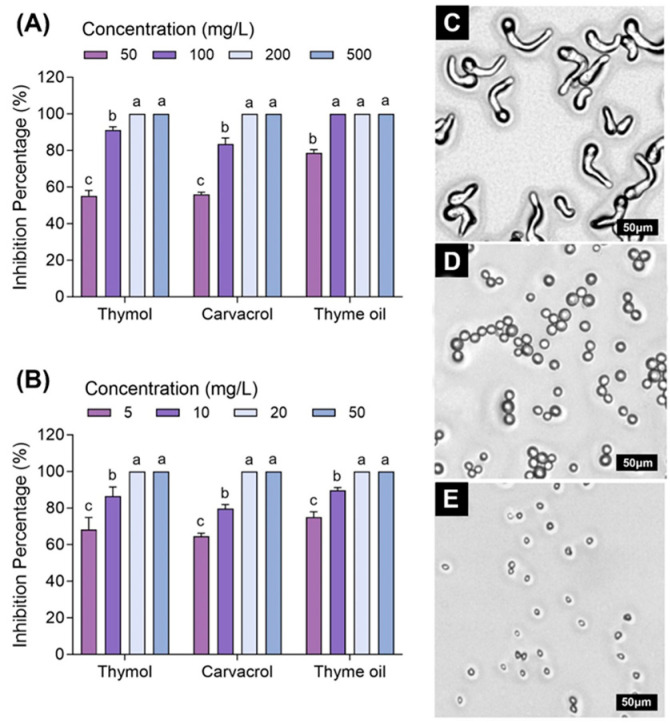
Effects of carvacrol, thymol, and thyme oil on *Rhizopus stolonifer* spore germination. Spore germination was observed at 5 h after inoculation with *R. stolonifer* spores at 22 ± 1 °C. (**A**) spore inhibition percentage by EOs in contact phase, (**B**) spore inhibition percentage by EOs in vapor phase, (**C**) control, (**D**) spore inhibited by 200 mg/L carvacrol in contact phase, (**E**) spore damaged by 50 mg/L thyme oil in vapor phase. The means followed by different letters at the same sampling point indicated statistical differences according to Tukey’s test (*p* < 0.05).

**Figure 4 foods-14-01273-f004:**
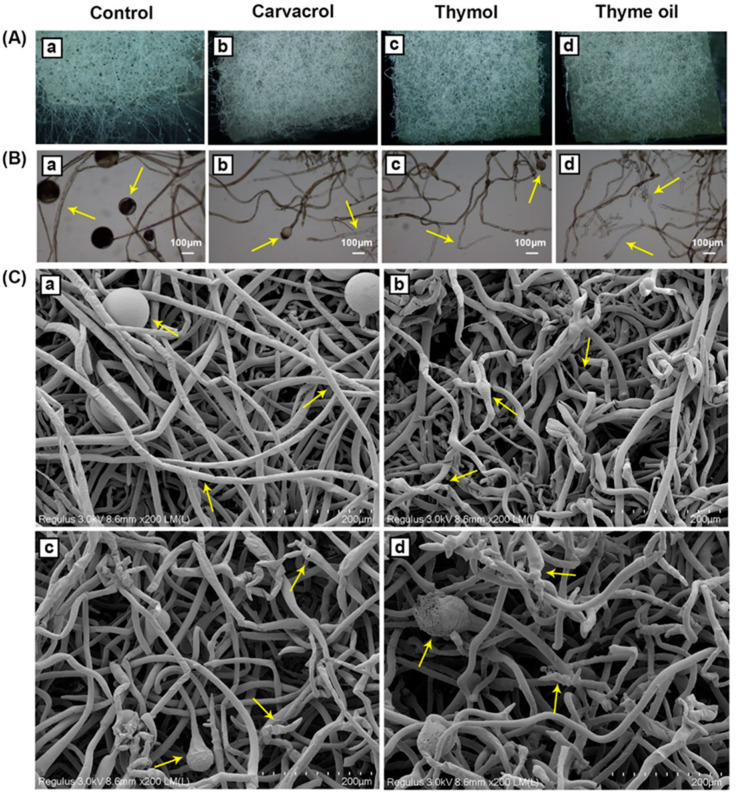
Effects of carvacrol, thymol, and thyme oil vapor treatments on *Rhizopus stolonifer* mycelial growth and morphological changes. Fungal mycelial growth on potato dextrose agar (PDA) after exposure to carvacrol, thymol and thyme oil (10 mg/L) for 12 h in a sealed container, fungal hyphal morphology under (**A**) stereomicroscope (10×), (**B**) microscope (40×, Bar = 100 μm) and (**C**) scanning electron microscopy observation (400×), alterations in hyphal shriveling and necrosis (arrows). (**a**) Untreated control, (**b**) Carvacrol vapored treatment, (**c**) Thymol vapored treatment, (**d**) Thyme oil vapored treatment.

**Figure 5 foods-14-01273-f005:**
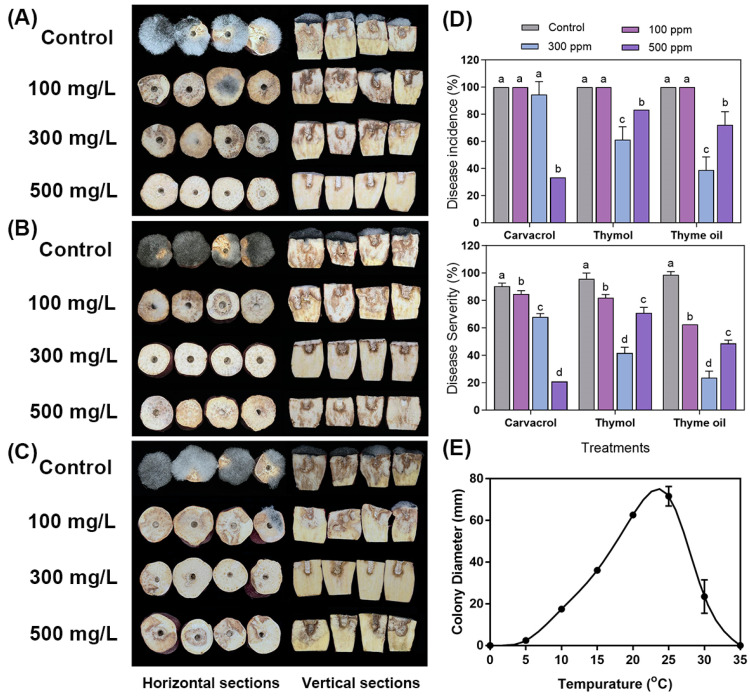
Effects of carvacrol, thymol, and thyme oil on *Rhizopus* soft rot control on sweet potato root sections inoculated with *R. stolonifer*. Symptoms and rot development on roots vaporized by (**A**) carvacrol, (**B**) thymol, and (**C**) thyme oil at various concentration levels and times. Disease incidence and severity (**D**) of carvacrol, thymol, and thyme oil observed at 60 h after fumigation at 100, 300, and 500 mg/L. (**E**) Effects of temperature on the mycelial growth of *Rhizopus stolonifer* in vitro. The means followed by different letters (a–d) at the same sampling point indicated statistical differences according to Tukey’s test (*p* < 0.05).

**Figure 6 foods-14-01273-f006:**
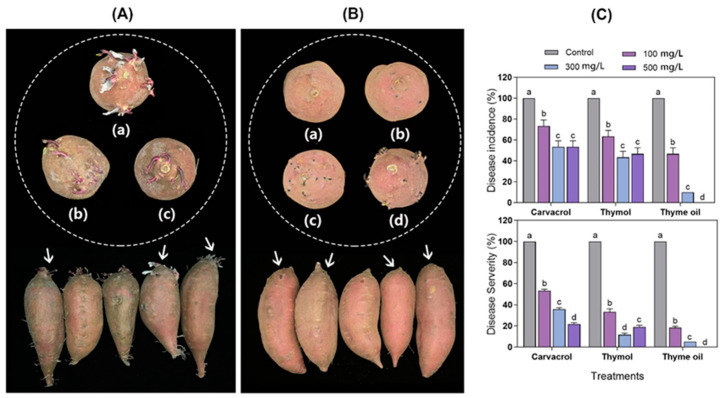
Effects of carvacrol, thymol, and thyme oil on *Rhizopus* soft rot control on non-artificially inoculated sweet potato roots. Symptoms of natural soft rot caused by *Rhizopus stolonifer* (**A**) Untreated control (**B**) roots vaporized by 300 mg/L thyme oil (**a**–**d**) the germination of control and thyme oil vapored treatment; (**C**) disease incidence and severity observed 21 d after essential oil treatments and incubation at 22 ± 1 °C. The means followed by different letters at the same sampling point indicated statistical differences according to Tukey’s test (*p* < 0.05).

**Figure 7 foods-14-01273-f007:**
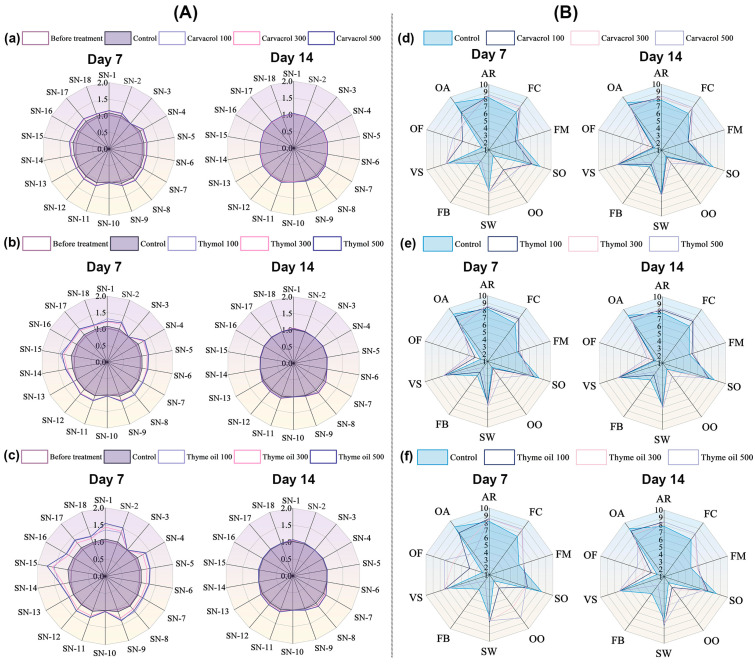
E-nose sensors (**A**) responding values of sweet potato flesh after essential oil treatments at 7 d and 14 d vaporized by (**a**) carvacrol, (**b**) thymol, and (**c**) thyme oil. Sensory analyses (**B**) of treated and untreated sweet potato roots vapored by (**d**) carvacrol, (**e**) thymol, and (**f**) thyme oil after 7 d of storage. Sensory attributes include the following: appearance (AR), flesh color (FC), firmness (FM), sweet potato odor (SO), off-odor (OO), sweetness (SW), fibrousness (FB), viscosity (VS), off-flavor (OF), and overall assessment (OA).

**Figure 8 foods-14-01273-f008:**
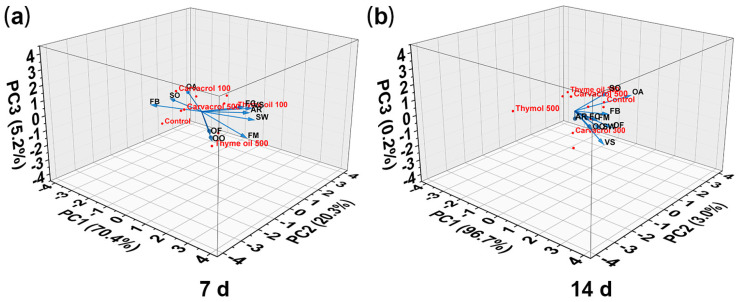
Three-dimensional principal component analysis (PCA) plot of sweet potato roots treated with carvacrol, thymol, and thyme oil compared to the control on (**a**) days 7 and (**b**) days 14.

**Table 1 foods-14-01273-t001:** The main constituents and average abundance of the volatiles in thyme oil.

Number	RT	Compounds	Average Abundance (Total Ion Current × 10^6^)
			Thyme Oil
1	21.679	α-Pinene	689.55
2	22.45	Camphene	209.71
3	23.732	β-Myrcene	376.51
4	24.694	β-Pinene	22.26
5	25.392	o-Cymene	35.70
6	25.709	p-Cymene	3207.05
7	25.819	D-Limonene	25.37
8	26.037	Eucalyptol	308.11
9	27.034	γ-Terpinene	1481.03
10	28.448	Linalool	1261.57
11	31.131	Camphor	533.46
12	31.44	Isoborneol	197.53
13	31.755	Borneol	315.01
14	32.047	4-Terpinenol	224.87
15	35.989	Thymol	5037.46
16	36.354	Carvacrol	932.21
17	36.692	2-(1-Methylpropyl)phenol	53.95
18	38.466	Eugenol	12.18
19	39.605	α-Copaene	20.39
20	41.356	Caryophyllene	561.03
21	42.459	Humulene	62.07
22	46.711	Caryophyllene oxide	155.99

**Table 2 foods-14-01273-t002:** Effects of thymol, carvacrol, and thyme oil on postharvest quality characteristics of sweet potato roots before and after 7 d storage at room conditions. Data are the mean of 24 roots ± standard deviation (SD). Data followed by the same letter within each column are not significantly different by the Duncan’s multiple range test (*p* < 0.05).

Treatments		Weight Loss (100%)	Firmness (N)	Starch (mg/g Fresh Weight)	Carotenoids (mg/100 g Fresh Weight)	Flavonoids (mg/100 g Fresh Weight)
Before storage			25.10 ± 0.92 ^a^	76.58 ± 2.31 ^b^	3.54 ± 0.12 ^b^	50.14 ± 1.67 ^a^
After storage						
Control		0.19 ± 0.02 ^a^	24.14 ± 0.44 ^a^	79.98 ± 2.48 ^ab^	3.66 ± 0.07 ^ab^	43.74 ± 2.51 ^b^
Carvacrol	100 mg/L	0.16 ± 0.01 ^b^	24.27 ± 0.30 ^a^	81.79 ± 4.01 ^ab^	3.66 ± 0.19 ^ab^	42.91 ± 3.79 ^b^
	300 mg/L	0.17 ± 0.01 ^ab^	24.31 ± 0.49 ^a^	80.12 ± 3.29 ^ab^	3.67 ± 0.13 ^ab^	42.76 ± 1.92 ^b^
	500 mg/L	0.17 ± 0.01 ^ab^	24.45 ± 0.50 ^a^	80.81 ± 2.69 ^ab^	3.67 ± 0.11 ^ab^	44.74 ± 2.05 ^b^
Thymol	100 mg/L	0.17 ± 0.01 ^ab^	24.37 ± 0.64 ^a^	81.28 ± 2.38 ^ab^	3.65 ± 0.24 ^ab^	42.69 ± 0.89 ^b^
	300 mg/L	0.17 ± 0.00 ^ab^	24.96 ± 0.48 ^a^	84.81 ± 1.96 ^a^	3.72 ± 0.12 ^ab^	45.08 ± 1.90 ^b^
	500 mg/L	0.17 ± 0.01 ^ab^	24.43 ± 0.63 ^a^	84.70 ± 2.81 ^a^	3.67 ± 0.27 ^ab^	43.62 ± 1.52 ^b^
Thyme oil	100 mg/L	0.17 ± 0.01 ^ab^	24.64 ± 0.45 ^a^	84.36 ± 2.44 ^a^	3.71 ± 0.14 ^ab^	44.81 ± 2.81 ^b^
	300 mg/L	0.17 ± 0.01 ^ab^	25.30 ± 0.51 ^a^	84.38 ± 3.29 ^a^	3.68 ± 0.10 ^ab^	44.56 ± 1.99 ^b^
	500 mg/L	0.17 ± 0.00 ^ab^	24.91 ± 1.01 ^a^	84.42 ± 2.37 ^a^	3.96 ± 0.10 ^a^	47.10 ± 3.73 ^ab^

**Table 3 foods-14-01273-t003:** Effects of thymol, carvacrol, and thyme oil on postharvest TPA parameters of sweet potato roots before and after 7 d storage at room conditions. Data are the mean of 24 roots ± standard deviation (SD). Data followed by the same letter within each column are not significantly different by Duncan’s multiple range test (*p* < 0.05).

Treatments		Adhesive Force(N)	Adhesiveness (mJ)	Cohesiveness (Ratio)	Springiness (mm)	Gumminess (N)	Chewiness (mj)
Before storage		−0.10 ± 0.00 ^b^	0.0072 ± 0.000 ^a^	0.68 ± 0.03 ^b^	1.39 ± 0.06 ^a^	103.41 ± 10.37 ^a^	131.55 ± 8.67 ^b^
After storage							
Control		−0.08 ± 0.00 ^ab^	0.0067 ± 0.000 ^a^	0.72 ± 0.02 ^ab^	1.36 ± 0.08 ^a^	96.56 ± 6.65 ^a^	144.43 ± 9.97 ^ab^
Carvacrol	100 mg/L	−0.08 ± 0.01 ^ab^	0.0073 ± 0.001 ^a^	0.76 ± 0.02 ^a^	1.36 ± 0.05 ^a^	103.33 ± 9.74 ^a^	142.60 ± 8.01 ^ab^
	300 mg/L	−0.08 ± 0.01 ^ab^	0.0066 ± 0.003 ^a^	0.75 ± 0.06 ^ab^	1.38 ± 0.05 ^a^	94.34 ± 8.16 ^a^	143.29 ± 11.67 ^ab^
	500 mg/L	−0.08 ± 0.01 ^ab^	0.0080 ± 0.000 ^a^	0.76 ± 0.02 ^a^	1.50 ± 0.10 ^a^	98.67 ± 4.16 ^a^	145.03 ± 11.35 ^ab^
Thymol	100 mg/L	−0.08 ± 0.02 ^ab^	0.0091 ± 0.000 ^a^	0.75 ± 0.03 ^ab^	1.39 ± 0.09 ^a^	105.28 ± 8.95 ^a^	143.56 ± 11.15 ^ab^
	300 mg/L	−0.06 ± 0.01 ^a^	0.0073 ± 0.001 ^a^	0.75 ± 0.03 ^ab^	1.37 ± 0.08 ^a^	104.56 ± 9.87 ^a^	143.71 ± 10.73 ^ab^
	500 mg/L	−0.08 ± 0.01 ^ab^	0.0090 ± 0.000 ^a^	0.75 ± 0.03 ^ab^	1.43 ± 0.11 ^a^	105.97 ± 7.76 ^a^	144.08 ± 8.48 ^ab^
Thyme oil	100 mg/L	−0.06 ± 0.01 ^a^	0.0066 ± 0.003 ^a^	0.75 ± 0.07 ^a^	1.43 ± 0.06 ^a^	94.00 ± 5.44 ^a^	145.39 ± 17.63 ^ab^
	300 mg/L	−0.07 ± 0.02 ^a^	0.0090 ± 0.000 ^a^	0.76 ± 0.04 ^a^	1.45 ± 0.08 ^a^	98.06 ± 7.48 ^a^	142.77 ± 4.44 ^ab^
	500 mg/L	−0.06 ± 0.01 ^a^	0.0088 ± 0.001 ^a^	0.73 ± 0.01 ^ab^	1.51 ± 0.07 ^a^	105.98 ± 8.12 ^a^	163.67 ± 12.75 ^a^

**Table 4 foods-14-01273-t004:** Effects of thymol, carvacrol, and thyme oil on postharvest color characteristics of sweet potato roots before and after 7 d storage at room conditions. Data are the mean of 24 roots ± standard deviation (SD). Data followed by the same letter within each column are not significantly different by Duncan’s multiple range test (*p* < 0.05).

Treatments		Hue		Chroma		Lightness (L)	
		Day 0	Day 7	Day 0	Day 7	Day 0	Day 7
Control		25.81 ± 1.54 ^bc^	29.96 ± 1.67 ^ab^	19.41 ± 0.24 ^b^	18.58 ± 1.15 ^b^	36.13 ± 0.87 ^a^	34.90 ± 2.46 ^abc^
Carvacrol	100 mg/L	25.36 ± 1.12 ^c^	29.15 ± 0.52 ^b^	22.66 ± 0.43 ^a^	21.81 ± 0.85 ^a^	35.47 ± 2.75 ^a^	33.03 ± 2.49 ^c^
	300 mg/L	26.60 ± 0.69 ^abc^	30.40 ± 1.36 ^ab^	20.29 ± 1.56 ^ab^	19.67 ± 1.29 ^ab^	38.03 ± 0.47 ^a^	36.63 ± 1.27 ^ab^
	500 mg/L	26.73 ± 0.80 ^abc^	30.85 ± 2.28 ^ab^	20.23 ± 1.51 ^ab^	19.71 ± 1.18 ^ab^	35.37 ± 1.86 ^a^	33.50 ± 1.21 ^abc^
Thymol	100 mg/L	27.06 ± 1.19 ^abc^	31.52 ± 0.75 ^ab^	21.46 ± 1.88 ^ab^	20.57 ± 0.79 ^ab^	37.47 ± 1.37 ^a^	35.37 ± 2.14 ^abc^
	300 mg/L	28.14 ± 1.60 ^ab^	32.51 ± 2.36 ^a^	19.48 ± 0.80 ^b^	18.61 ± 0.66 ^b^	37.67 ± 2.53 ^a^	35.33 ± 0.49 ^abc^
	500 mg/L	27.17 ± 1.94 ^abc^	31.03 ± 1.54 ^ab^	21.01 ± 1.64 ^ab^	20.68 ± 1.96 ^ab^	38.87 ± 2.90 ^a^	36.10 ± 1.77 ^abc^
Thyme oil	100 mg/L	25.32 ± 1.10 ^c^	29.92 ± 1.51 ^ab^	21.59 ± 1.98 ^ab^	20.44 ± 0.90 ^ab^	36.50 ± 0.35 ^a^	34.43 ± 1.46 ^abc^
	300 mg/L	27.82 ± 1.26 ^ab^	31.40 ± 1.45 ^ab^	21.81 ± 0.66 ^ab^	20.47 ± 1.92 ^ab^	35.73 ± 1.36 ^a^	33.43 ± 0.68 ^abc^
	500 mg/L	28.37 ± 0.75 ^a^	32.13 ± 1.92 ^ab^	22.26 ± 1.19 ^a^	21.16 ± 0.49 ^a^	38.47 ± 1.80 ^a^	36.77 ± 1.32 ^a^

## Data Availability

The original contributions presented in this study are included in the article/Appendix A. Further inquiries can be directed to the corresponding author.

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
