# Peer review of "The Control of Postharvest Soft Rot Caused by Rhizopus stolonifer on Kokei No. 14 Organic Sweet Potato Roots by Carvacrol, Thymol, and Thyme Oil"

_foods, 2025, doi:10.3390/foods14071273_

Round 1
Reviewer 1 Report
Comments and Suggestions for Authors
The work is significant. My comments are in the file attached.

Reviewer 2 Report
Comments and Suggestions for Authors
The manuscript 3516602 was reviewed, which focuses on controlling a fungal species using essential oils. Although not a new topic, the manuscript provides abundant information that supports the results and thus contributes to the use of essential oils in the postharvest handling of fruits and vegetables. The following are some aspect to be considered:
Check if it is correct to name carvacrol and thymol as essential oils. Both are rather terpenic compounds, which are present in many essential oils, for example in thyme essential oil as it was confirmed. This observation applies to all the manuscript.
The molecular method mentioned in section 2.1 that was used to identify the pathogenic species must be described and results must be included in the Results section.
The results of the experiment described in section 2.3 show that lower concentrations of the inhibitors in the volatilized mode produce the same effect as the higher concentrations used in the direct contact method (Figure 1). In this regard, it is appropriate to declare a minimum inhibitory concentration (MIC) in each case. According to Figure 1, the MIC was between 5 and 10 mg/L for the three inhibitors in the vapor contact method. This was consistent with the experiment described in section 2.4, where 10 mg/L was used. However, why were these results not used in the experiments described in sections 2.5 and 2.6, where concentrations were like those used in the contact method?
How was the actual concentration of the inhibitors in the pathogen's headspace determined in the vapor exposure method? Due to their high boiling point, essential oils have low volatility, which means that the actual concentrations were likely lower than indicated.
In general, manuscript provides sufficient information to confirm the inhibitory effect of thyme essential oil and the terpene compounds thymol and carvacrol.
It is advisable to update many of the references.
Round 2
Reviewer 1 Report
Comments and Suggestions for Authors
I am impressed by the revised manuscript. I recommend acceptance